# Pediatric Infections by *Human mastadenovirus C* Types 2, 89, and a Recombinant Type Detected in Japan between 2011 and 2018

**DOI:** 10.3390/v11121131

**Published:** 2019-12-06

**Authors:** Kenichiro Takahashi, Gabriel Gonzalez, Masaaki Kobayashi, Nozomu Hanaoka, Michael J. Carr, Masami Konagaya, Naomi Nojiri, Miki Ogi, Tsuguto Fujimoto

**Affiliations:** 1Division 4, Infectious Disease Surveillance Center, National Institute of Infectious Diseases, Tokyo 162-8640, Japan; taka_ken1129@yahoo.co.jp (K.T.); nozomu@nih.go.jp (N.H.); gaya@nih.go.jp (M.K.); nojinoji@nih.go.jp (N.N.); 2Department of Pediatrics, Tokyo Women’s Medical University Medical Center East, Tokyo 116-8567, Japan; 3Research Center for Zoonosis Control, Hokkaido University, Sapporo 001-0020, Japan; gagonzalez@czc.hokudai.ac.jp; 4Kobayashi Pediatric Clinic, Fujieda 426-0067, Japan; koba-m@if-n.ne.jp; 5National Virus Reference Laboratory, School of Medicine, University College Dublin, Dublin D04 V1W8, Ireland; michael.carr@ucd.ie; 6Global Station for Zoonosis Control, Global Institution for Collaborative Research and Education (GI-CoRE), Hokkaido University, Sapporo 001-0020, Japan; 7Infectious Disease Research Division, Hyogo Prefectural Institute of Public Health Science, Kakogawa 675-0003, Japan; Miki_Ogi@pref.hyogo.lg.jp

**Keywords:** *Human mastadenovirus C*, respiratory infections, pediatric infections, recombination, molecular epidemiology, adenovirus typing

## Abstract

Between 2011 and 2018, 518 respiratory adenovirus infections were diagnosed in a pediatric clinic in Shizuoka, Japan. Detection and typing were performed by partial sequencing of both hexon- and fiber-coding regions which identified: adenovirus type 1 (Ad-1, *n* = 85), Ad-2 (*n* = 160), Ad-3 (*n* = 193), Ad-4 (*n* = 18), Ad-5 (*n* = 27), Ad-11 (*n* = 2), Ad-54 (*n* = 3), and Ad-56 (*n* = 1). Considering previous reports of the circulation of an endemic recombinant Ad-2, e.g., Ad-89, 100 samples typed as Ad-2 were randomly selected for further molecular typing by sequencing the penton base-coding region. Despite the high nucleotide sequence conservation in the penton base- coding region, 27 samples showed 98% identity to Ad-2. Furthermore, 14 samples showed 97.7% identity to Ad-2 and 99.8% identity to Ad-89, while the remaining 13 samples showed an average 98% pairwise identity to other Ad-C types and clustered with Ad-5. The samples typed as Ad-89 (*n* = 14) and as a recombinant Ad type (P5H2F2) (*n* = 13) represented 27% of cases originally diagnosed as Ad-2, and were detected sporadically. Therefore, two previously uncharacterized types in Japan, Ad-89 and a recombinant Ad-C, were shown to circulate in children. This study creates a precedent to evaluate the epidemiology and divergence among Ad-C types by comprehensively considering the type classification of adenoviruses.

## 1. Introduction

Members of the *Adenoviridae* family are non-enveloped double-stranded DNA viruses with a size ranging from 70 to 90 nm. Among them, human mastadenoviruses (Ad) infect humans and cause a variety of diseases [1,2]. Human mastadenoviruses are classified into seven species from Ad-A to Ad-G, according to a variety of virological, clinical, and phylogenetic characteristics [1]. *Human mastadenovirus C* (Ad-C) types are associated with ~5% of upper respiratory tract infections and ~15% of lower respiratory infections in children under the age of five [3,4]. Among Ad-C, adenovirus type 2 (Ad-2) is one of the most commonly detected types worldwide and is a leading pathogen associated with pediatric respiratory tract infections. Ad-C types are harbored as latent infections in tonsillar tissue and, also are intermittently excreted in the feces of children [5]. It is noteworthy that despite the rarity in frequency, adenovirus encephalitis due to Ad-2 infections has been reported in immunocompromised patients [6]. Ad-2 ranks as the most frequently detected type in the Japanese surveillance of adenovirus infections [7], and a similar trend is also observed in the United States of America [8]. The proper identification and classification of adenovirus types and outbreaks are an important part of the surveillance of infectious agents performed by the National Institute of Infectious Diseases of Japan [9].

Since 2007, molecular analyses have increasingly supplanted serological testing as adenovirus typing methodologies, after the recognition of adenovirus type 52 and a novel species, Ad-G, based on genomic analysis [10]. Whole genome sequence analysis has become increasingly common for the complete characterization of adenovirus types [1,11], effectively replacing serotyping due to the replicability and convenience of molecular techniques [12]. In addition, genomic analysis allows consideration of adenovirus properties largely overlooked by serotyping methods, such as recombination events affecting single or multiple epitope-determining regions [13,14]. The shift in typing methodologies has consequently favored genotypic assignment based on whole genome sequence instead of serological analyses, and for simplicity hereafter “type” will refer to both. Furthermore, the taxonomy of adenoviruses can present issues as alternative classification schemes based solely on either genotypic or serological criteria are inadequate to accommodate all biological and clinical characteristics of adenoviral species [15]. However, revisions to virus taxonomy should be embraced to better reflect the current understanding of viral evolution and the progress of genomics technology [12]. Moreover, within Ad-C, type 57 (Ad-57) was reported in Russia and China [16] in 2011, and type 89 (Ad-89) was reported in Germany in 2019 [17]. The circulation of Ad-C57 in Japan has also been reported [18]; however, the circulation of Ad-89 remains to be elucidated.

Despite Ad-89 and Ad-2 sharing high pairwise similarity in hexon- and fiber-coding regions, Ad-89 was initially reported as a novel type based on the criterion of a unique sequence in the penton base-coding region distinguishable from other Ad-C types [17]. The penton base protein, one of the major capsid proteins, elicits part of the immune response to Ad infection; therefore, the information on the penton base region is also considered for type characterization [19,20]. Although the criteria for typing is based on the three major capsid proteins [11], in routine practice, detection and typing methodologies focus solely on the hexon region, and to a lesser degree on fiber, and largely neglect the penton base region, which may create biases and misidentifications within Ad-C types. Therefore, a large amount of collected sequence information from Ad-2 from diverse samples tends to be insufficient for proper typing, due to the overreliance on the hexon-coding region as the main determinant of the type while neglecting other genomic regions [21].

Considering that Ad-2 is the major source of respiratory infections in Japan, representing 26% of diagnosed adenovirus infections between 2008 and 2017 [9], and the previously reported evidence of recombinant forms of serotype Ad-2, we identified that the circulation of such recombinant types in Japan remains to be elucidated. Therefore, the present study assessed Ad-2 strains that could potentially have been mistyped considering discordances in type assignment based on penton base-, hexon-, and fiber-coding regions. We hypothesized that a percentage of the circulating novel types has been detected previously but misclassified over time due to the prevailing typing methodologies focusing predominantly on the hexon region [22,23]. This study retrospectively examined 100 samples originally typed as Ad-2 from 518 adenovirus respiratory infections in children attending a pediatric clinic in Japan between 2011 and 2018. The viral sequences and the clinical data of these infections were further analyzed confirming the circulation of Ad-89 in Japan and the identification of a recombinant Ad-C type.

## 2. Materials and Methods

### 2.1. Patients, Clinical Data, and Adenovirus Sample Collection

Pharyngeal swabs from respiratory infections were collected with informed consent between 2011 and 2018 in Kobayashi Pediatric Clinic, Shizuoka prefecture, Japan. The following demographic and clinical data were collected for adenovirus-positive patients: (1) examination date, (2) age and gender, (3) clinical manifestations (maximum temperature, duration of fever, and conjunctival injection and/or ocular symptoms), (4) laboratory tests (white blood cell count (WBC) and C-reactive protein, (CRP)), and (5) clinical diagnosis (lower respiratory tract infection, and exudative tonsillitis with infected pharyngeal infection and tonsillar exudate). Lower respiratory tract infections were diagnosed by the physician based on auscultation findings and clinical observations.

To check for differences in clinical data between groups, clinical data were statistically evaluated using Fisher’s exact test for dichotomous variables (gender, ocular symptoms, exudative tonsillitis, and lower respiratory tract infection) and Kruskal–Wallis test for continuous variables (age, maximum temperature, duration of fever, WBC, and CRP). A *p*-value < 0.05 was considered as significant. All statistical analyses were performed using JMP 14 (SAS Institute Inc., Cary, NC, USA).

Pharyngeal specimens were collected from the pharynx of pediatric patients suspected of adenovirus respiratory tract infections and were tested as positive for adenovirus in screening tests with immunochromatographic kits [7]. The viral genomes were extracted from the throat swabs using the High Pure Viral Nucleic Acid Kit (Roche, Basel, Switzerland). The number of adenovirus genome copies in clinical specimens was determined by real-time PCR, as reported previously [24].

### 2.2. Determination of Partial Sequences of Hexon Loop and Penton Base-Coding Regions

The nucleotide sequence corresponding to the hexon loop 1 was sequenced (707–710 bp) from all samples, following a nested PCR, as reported previously [25]. The size of amplified PCR products was confirmed by 1.5% (*w*/*v*) agarose gel electrophoresis, and amplicons of the predicted size were excised from the gel and purified by the FastGene Gel/PCR extraction kit (Nippon genetics, Tokyo, Japan). The nucleotide sequences of the purified PCR products were determined by Sanger sequencing using standard approaches. The sequences were compared to other known sequences deposited in the International Nucleotide Sequence Database Consortium (INSDC) by BLAST. The partial nucleotide sequence corresponding to the penton base (~1140 bp), including the tripeptide Arg-Gly-Asp (RGD) loop coding region, was compared as described in a previous report [26]. The partial sequences of penton base- and hexon-coding regions were deposited in the DNA Data Bank of Japan (DDBJ), part of the INSDC, with the accession numbers LC498883-LC498982 and LC499882-LC499981, respectively.

### 2.3. Multiplex PCR Using the Fiber Region

A multiplex-PCR that simultaneously targets Ad-1, -2, -5, -6, and -57 was employed to amplify the unique sequences within the fiber-coding regions and discriminated types based on amplicon size determined by agarose gel electrophoresis [7]. If multiple bands were detected for a single sample at the expected sizes, the sample was considered a co-infection. It is noteworthy that Ad-2, Ad-89, and the potential recombinant type described in this study share almost identical fiber-coding sequences and are consequently indistinguishable by this methodology.

### 2.4. Virus Isolation and Neutralization

HAdVs were isolated using the A549 cell line, as previously described [7]. There was a total of six isolates, which were chosen to include differentiable penton base regions and tested by neutralization analysis using rabbit antiserum against Ad-1, Ad-2, Ad-5, and Ad-6 (Denka-seiken, Tokyo, Japan). Neutralization reactions were performed with the antisera at concentrations of 5 U, 10 U, and 20 U, using the viral solution as the attack virus at concentrations at which approximately 75% of cytopathic effect (CPE) on A549 cells were observed after 48 h.

### 2.5. Whole Genome Sequence Analysis

Isolated viruses were extracted from the virus culture solutions and the whole genome sequences were determined by next generation sequencing on the NovaSeq 6000 platform (Illumina, San Diego, CA, USA). Assemblies were performed with CLC Genomics Workbench 12 (QIAGEN, Tokyo, Japan). Also, the inverted terminal repeats (ITR) of both genome sequences were determined by Sanger methods. The complete genome sequences were annotated and deposited in DDBJ under the accession numbers LC504573 for the Ad-89 and LC504572 for the recombinant Ad-P5H2F2 strains.

### 2.6. Sequence Analysis

Partial and complete genome sequences of samples were compared to reference sequences retrieved from the INSDC. The accession numbers of the reference sequences were: Ad-1 (AC000017), Ad-2 (AC000007, MF044052), Ad-5 (AC000008), Ad-6 (FJ349096), Ad-57 (HQ003817), Ad-89 (MH121097, MH121114). Ad-B3 (AY599834) was used as an outgroup to root trees. Multiple sequence alignments (MSAs) including obtained sequences and the reference sequences of Ad-C were built with MAFFT [27] and the corresponding phylogenies were inferred with MrBayes v3.5 [28]. The best models were chosen by MSA using the corrected Akaike information criterion (AIC) calculated by MEGA (version 7) [29] and the Bayesian chain length was 2 × 10^6^ to assure convergence. Recombination analysis was performed using SimPlot (version 3.5.1) [30] (with window and step sizes as 500 bp and 100 bp, respectively, and 100 repetitions) and the recombination detection program (RDP) v4.0 [31], setting the program to consider reliable recombination events with a significance of *p* < 0.01 in at least three out of seven selected methods: RDP, GENECONV, BootScan, Maxchi, Chimaera, SiSscan, and 3Seq.

### 2.7. Ethical Considerations

The ethical review of this study was approved by the Ethical Review Board of the National Institute of Infectious Diseases (No. 877, 13 February, 2018), and the Japan Medical Association Ethical Review Board (No.30-6, 19 February, 2019).

## 3. Results

### 3.1. Adenovirus Detection and Typing by Hexon- and Fiber-Coding Regions

Between 2011 and 2018, a pediatric clinic in Shizuoka diagnosed 518 pediatric patients as adenoviral respiratory tract infections using immunochromatography kits. The adenoviruses were both detected and typed via PCR in 489 samples (94% of cases). The hexon partial sequences of such samples were submitted by BLAST to the INSDC and determined the infecting types as: Ad-1 (*n* = 85), Ad-2 (160), Ad-3 (193), Ad-4 (18), Ad-5 (27), Ad-11 (2), Ad-54 (3), and Ad-56 (1), which were distributed in relatively comparable frequencies across the study years (Figure 1). These classifications were concordant with the multiplex PCR results targeting the fiber-coding region of the adenoviral genomes.

### 3.2. Nucleotide Sequencing of Penton Base- and Hexon-Coding Regions from Ad-2 Positive Samples

The adenovirus copy number was measured in 160 specimens typed as Ad-2; among them, 100 specimens were randomly selected from specimens that contained >10^4^ genome copies/μL. The hexon and penton coding regions of these 100 samples were sequenced. The naming of these strains was in the format K#-XXX-isolation year, from K1-006-2012 to K100-390-2016, as shown in Figure 2 (Appendix A).

The lengths of the 100 partial sequences targeting the penton base encompassing the RGD loop were 1140–1152 bp (yielding a consensus length of 1152 bp) and the lengths of sequences covering the hexon loop 1 coding regions obtained were 707–710 bp (an 896-bp consensus length). The phylogenetic trees for penton base and hexon, considering these and other Ad-C reference sequences, were inferred with MrBayes using the general time reversible nucleotide substitution model allowing for heterogeneity modeled with a gamma distribution (GTR+Γ) (Figure 2A,B). In the penton base phylogenetic tree (Figure 2A), 14 specimens (14%) formed a highly-supported cluster with two strains previously reported as Ad-89 from Germany and shared high nucleotide sequence similarity (0.99 ± 0.002); therefore, these specimens were reclassified as Ad-89. Interestingly, 13 samples (13%) formed a new previously unrecognized cluster that was distinct from Ad-2 (0.975 pairwise identity), Ad-89 (0.983 pairwise identity), and Ad-5 (0.981 pairwise identity). The remaining 73 samples clustered with high support to previously described clusters containing Ad-1, Ad-57, and Ad-6, with Ad-2 as an outgroup (Figure 2A). Furthermore, the similarity of these samples to Ad-1 and Ad-2 was 0.998 + 0.0009 and 0.988 ± 0.0008, respectively. These findings were consistent with the interpretation that the penton base-coding region has been affected by multiple recombination events during Ad evolution (Appendix A).

The phylogenetic tree corresponding to the loop 1 of the hexon region clustered all samples from this study with Ad-2 reference sequences (Figure 2B). Despite the contrast in similarity to other Ad-C types (0.774 ± 0.009 average), Ad-89 samples evidenced a distinguishable difference to Ad-2 (0.98 ± 0.01 average). Moreover, the group of sequences identified as a recombinant type, based on the penton base, showed high similarity in the hexon loop 1 to Ad-2 (0.995 ± 0.003 average) (Appendix A). The phylogenetic results were further confirmed by neutralization tests by antisera against Ad-2 (Table 1).

### 3.3. Complete Genome Sequences for Ad-89 and a Putatively Recombinant Ad Type

The complete genome sequences of two samples were determined to definitively assign the viral type based on genomic analysis. The sequence of K67-339-2016, classified as Ad-89, and K19-85-2012, representing a putative recombinant type, were obtained by next-generation sequencing and assembled from over 20M reads in each sample. The genomic terminal regions were confirmed by Sanger methods for both samples. The assembled genomes of Ad-89 K67-339-2016 and K19-85-2012 were 35,911 bp and 35,921 bp in length, respectively. The nucleotide composition of both sequences showed the characteristic 55% content of guanine and cytosine (GC%) from Ad-C; however, K19-85-2012 derived from the putative recombinant Ad type showed the lowest GC% among Ad-C types with 55.15%, while other types showed an average 55.24% ± 0.06. The annotation of both genomes showed the expected gene order for Ad-C members.

The phylogenetic tree of the genome sequences with other reference Ad-C sequences confirmed the classification of K67-339-2016 as Ad-89 at the genome level (Figure 3A). However, the highly supported position of K19-85-2012 was an outgroup to Ad-2 and Ad-89. Furthermore, the phylogenetic trees corresponding to the coding sequences of penton base, hexon, and fiber (Figure 3B–D) clustered K19-85-2012 with Ad-5, Ad-2, and Ad-2, respectively, summarized as P5/H2/F2.

The discordances in type assignment based on the major capsid protein coding regions suggested the possibility of recombination events which we explored further with the RDP program identifying four putative recombination events (Table 2). The impact of these recombination events was assessed further by BootScan and a similarity analyses in Simplot (Figure 4). Notably, the major differences to other Ad types were located in the hexon- and fiber-coding regions, in support of methodological approaches assessing both these sections of the Ad genome for type assignment. The BootScan analysis supported a chimeric origin for K19-85-2012 (Figure 4A) that pointed to regions coding the hexon and fiber as possibly recombinant areas and explaining their clustering with Ad-2. Analogously, the similarity sliding window analysis shows high conservation in the penton base- coding region with small variations that could explain K19-85-2012 clustering to Ad-5 (Figure 4D). The effects of the recombination events deduced from RDP and Simplot were further assessed with phylogenies considering the concatenation of sections detected as not recombinant (Figure 5A) and sections of the four detected putative recombination events (Figure 5B–E). Interestingly, the major part of K19-85-2012 clustered with Ad-5 (Figure 5A), while events 1 and 3 explained the clustering with Ad-2 (Figure 5B,D). Event 2 suggested the recombinant parent of the E3 region was Ad-57 (see Figure 5C).

### 3.4. Clinical Data

Out of the 489 adenovirus-positive cases from respiratory samples, 27 cases were excluded because of insufficient clinical information. The remaining cases comprised 270 boys and 192 girls with a median age of 33 months (interquartile range: 18–57.5 months). Among these cases, 155 cases presented exudative tonsillitis and 126 cases involved lower respiratory tract infections. Ninety-seven of the 100 Ad-2 cases selected for molecular characterization were further analyzed as sufficient clinical information was collected. The distribution of the causative agent of such cases was: Ad-89 (13 cases), Ad-P5H2F2 (13 cases), and Ad-2 (71 cases). The statistical analysis showed no evidence that Ad-2, Ad-89, and the putatively recombinant type had any noticeable difference in any of the recorded clinical variables such as age, maximum temperature, duration of fever, ocular symptoms, WBC, CRP, exudative tonsillitis, or lower respiratory tract infection (Appendix A).

## 4. Discussion

The current retrospective study characterized samples taken from pediatric respiratory infections over the 2011–2018 period and originally typed with cognizance to the common adenovirus types frequently diagnosed in Japan. Notably, more detailed characterization of samples typed as Ad-2 led to the uncovering of two recombinant forms involving serotype Ad-2 previously uncharacterized in Japan. The most common method of adenovirus detection in infections uses immunochromatography kits [24]; however, such testing only detects the presence of adenovirus and lacks the ability to provide further information about the viral type. Despite molecular approaches such as partial nucleotide sequencing of sub-genomic regions, which can provide further details on the identity of the infectious agents, the selection of the genomic regions to sequence or analyze can lead to misclassification [21]. The results of this study showed that a relatively high percentage (27%) of the Ad-2 samples in circulation during the years 2011–2018 were mistyped due to the overreliance on a classification scheme based exclusively on the hexon-coding region; furthermore, serological tests on the samples also showed neutralization reactions by rabbit serum against Ad-2. Therefore, although the available evidence shows that these variants have recombinant origins and have circulated in Japan for at least the previous decade, they have remained undetected by routine molecular typing and serological approaches.

The criteria for type classification considers analysis of the phylogenetic clustering of types in the three major capsid proteins penton base, hexon, and fiber, which acknowledges the independent variation that arises in their coding regions during viral evolution [11]. The adaptive immune response against adenovirus involves developing antibodies against each of the three major capsid proteins. Nevertheless, due to the surface occupied by each kind of protein in the capsid, the major part of the antibody production is focused against the hexon protein, followed by the fiber and the penton base [19,20,32]. Therefore, despite the major antigenic response residing within the hexon, the synergistic neutralization of fiber and penton base proteins play an important role in the complete neutralization of the virus [32]. Additionally, the adenovirus type criteria recognizes the tendency of Ad to recombine [11,13]. As this criteria considers three different and non-contiguous coding regions in the adenovirus genome, the criteria also allow assessment of the putative recombination events in such regions (Figure 4C).

Adenovirus is a common source of respiratory infections among children [33]. Ad-2 and *Human mastadenovirus B* type 3 (Ad-3) are among the most frequently diagnosed agents reported on an annual basis [7,34]. The properties of such types leading to their frequent detection in infectious cases remain to be explained. The available evidence points to the possibility of recombinant variants being mistyped as Ad-2 due to a similar hexon-coding region, and, as a consequence, affecting classification based on serological results. It is noteworthy that frequent illegitimate recombination of the hexon coding region in Ad-C was recognized even prior to the availability of the current number of genomic sequences [14]. In addition, considering the effects and frequency of recombination affecting the coding regions corresponding to epitope determinants in other adenoviral species, penton base, hexon, and fiber have been reported as recombination hotspots [13,35,36]; however, the mechanism(s) favoring the recombination at these loci remains to be elucidated.

Our results showed a potential caveat of existing typing approaches based solely on partial sequencing of hexon-coding regions, as such methodologies are not appropriate for detection of Ad-89 or the recombinant Ad-P5H2F2 described in the present study. These samples were also shown to be distinguishable from Ad-2 based on the penton base. In consideration of these results, International Nucleotide Sequence Database Collaboration (INSDC) was examined with BLAST queries for strains which could be considered Ad-89 or Ad-P5H2F2 but registered as Ad-2 (as of 11 November 2019). The results suggested multiple sequences classified as Ad-2 but with high similarity to Ad-89 (percent identity >99%) in the United States (KX384959, KF268130), Argentina (JX173079, JX173077.1), and from the original reports in Germany (MH121097, MH121114); hence, although the number of available Ad-89 sequences is small, the diversity of geographical origin of the sequences suggests a worldwide circulation of the type. BLAST results for the Ad-P5H2F2 returned only sequences with percent identities lower than 98.5%, supporting the novelty of the characterized sequence.

Ad-C is an endemic source of infection in all geographic regions, and circulates continuously [24,37]. However, the most prevalent type varies between different geographic locations over time likely attributable to the numbers of immunologically naïve individuals present in a population to maintain transmission. The present study focused on a single clinic in central Japan; consequently, it highlights the need to explore other Japanese locations and other countries to generate a more accurate epidemiological characterization of factors associated with the spread of Ad-89 and other Ad-C types and their differential clinical presentation in pediatric and adult cohorts. It is noteworthy that although Ad-C infections can cause persistent infections with viral shedding that could limit the determination of a definitive diagnosis of the pathogen eliciting the exhibited symptoms [38], the association of symptoms upon clinical presentation and the Ad types were supported by three factors: (1) the symptoms were summarized from multiple cases; (2) other types of Ads were not detected in the infections; and (3) the high concentrations of viral particles per sample needed to be detectable by immunochromatography kits suggests active infections.

In comparison with types in other Ad species, respiratory infection by Ad-C types has been reported to have higher levels of inflammatory cytokines such as G-CSF, IL-6, and TNF-α, and WBC [7]; however, no significant differences were found among Ad-C types. Analogously, the present study found no significant clinical differences among cases associated with types Ad-2, Ad-89, and Ad-P5H2F2 infections. It remains to be determined whether the recombinant origins and genomic divergence characterized in Ad-89 and Ad-P5H2F2 are reflected in the severity, duration, infectivity, or clinical manifestations in relation to Ad-2 infections, as has been suggested in other adenovirus species with frequent recombination events [39,40]. Moreover, considering the role of the penton base in the internalization of the virion by the host cell [41], future research should assess the effects of these recombination events among types in the internalization process. It also remains to be explored whether infections by these recombinant types afford any protection against Ad-2 and Ad-5 infection.

## 5. Conclusions

In the current study, we assessed the percentage of infections mistyped as Ad-2 due to typing methodologies based solely on the hexon region by further characterization of coding regions corresponding to the penton base. Twenty-seven percent of samples were shown to belong to a group distinguishable from Ad-2 based on the penton base; moreover, 13% of samples were characterized as a potentially novel Ad type based on recombination events. Additional studies including these recombinant variants will be useful to elucidate whether clinical differences can be attributed to their recombinant origins. This study provides insights on the pitfalls of currently practiced molecular typing methodologies and the divergence among Ad-C types, and our data underscore the importance of detailed molecular epidemiological assessment employing each of the major capsid genes and downstream genomic analyses to uncover further recombinant types. Two previously uncharacterized types in Japan, Ad-89 and a recombinant Ad-C type, were shown to circulate in children. This study creates a precedent to evaluate the epidemiology and divergence among Ad-C types by comprehensively considering the type classification of adenoviruses.

## Figures and Tables

**Figure 1 viruses-11-01131-f001:**
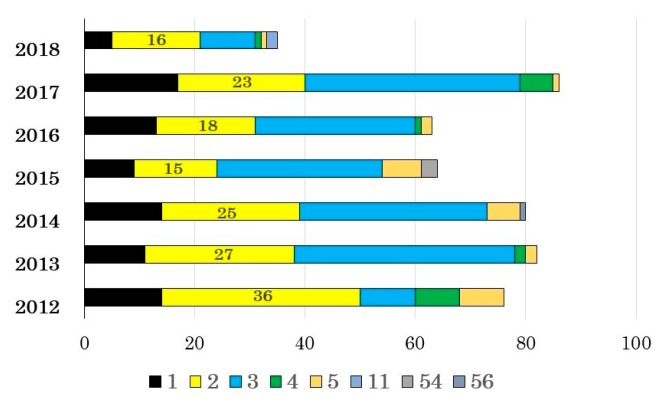
Adenoviruses (Ads) detected from respiratory patients during 2012 to 2018. The vertical axis shows the year, and the horizontal axis shows the number of Ads detected. Data from 2011 were not shown because only three strains were detected in 2011, Ad-1 (*n* = 2) and Ad-3 (*n* = 1).

**Figure 2 viruses-11-01131-f002:**
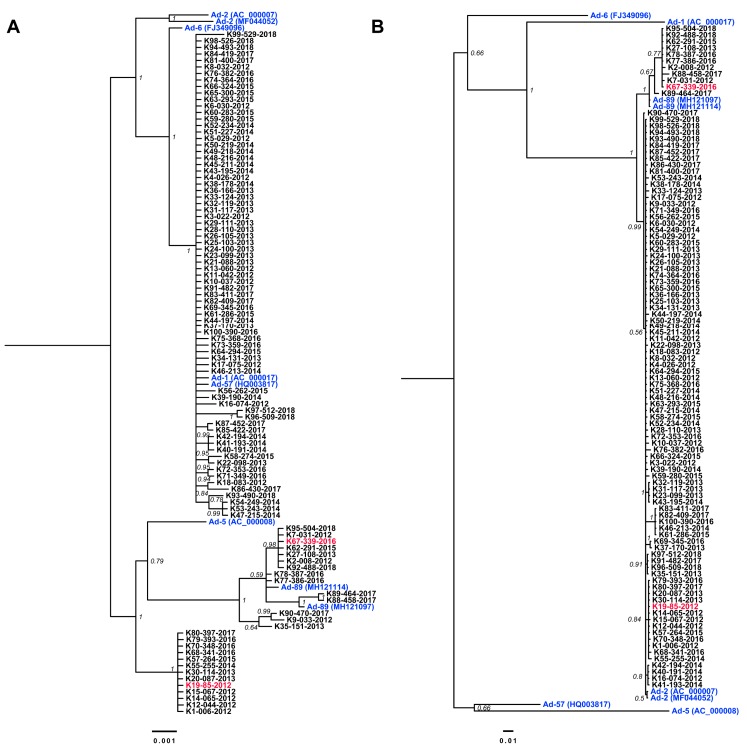
Phylogenetic trees of (**A**) the penton base RGD loop and (**B**) hexon loop 1. The posterior probability supporting the branching is shown at the tree nodes. Tip names are colored black for sequences obtained in the present study, blue for Ad-C reference sequences, and red for two samples used for whole-genome sequencing.

**Figure 3 viruses-11-01131-f003:**
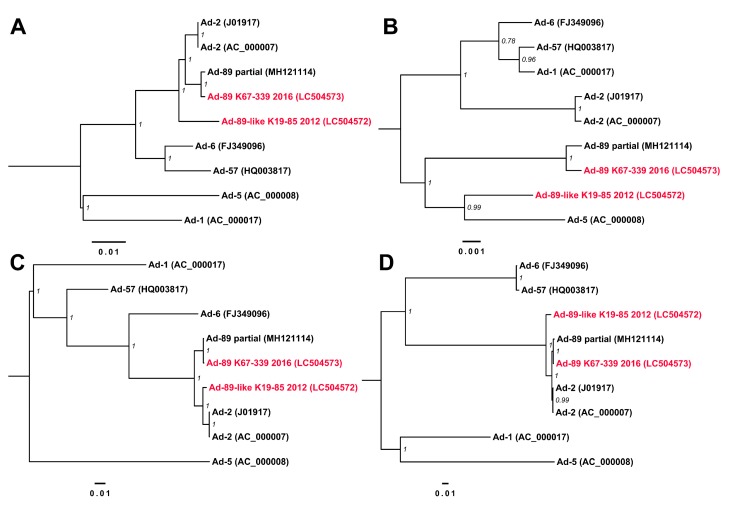
Phylogenetic trees for Ad-C types. The phylogenetic trees were inferred for (**A**) the complete genome sequences, and (**B**) penton base-, (**C**) hexon-, and (**D**) fiber-coding regions. The Bayesian posterior probability supporting the branching is shown adjacent to the nodes. Tip names of the recombinant genome sequences from the present study are colored in red.

**Figure 4 viruses-11-01131-f004:**
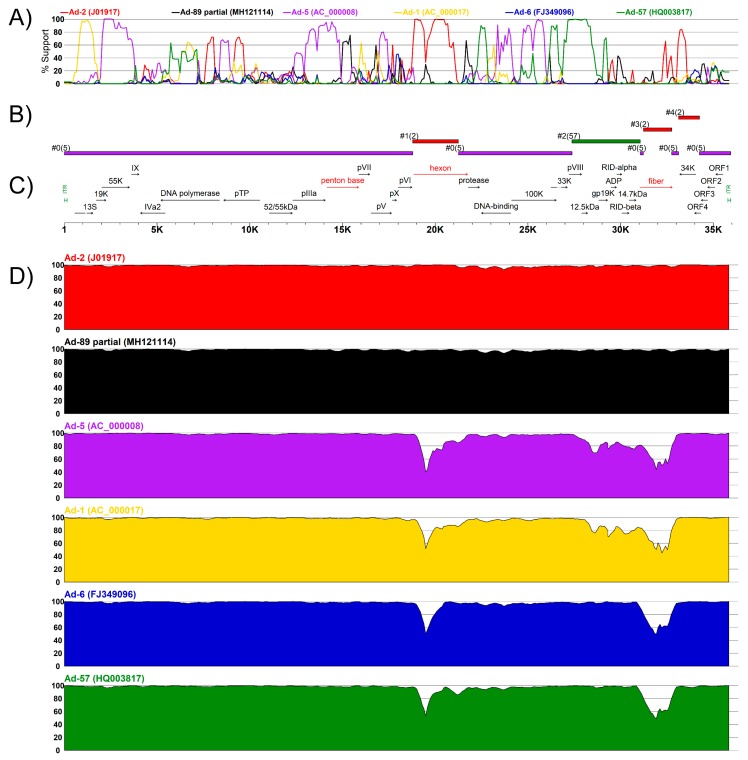
Recombination analysis by sliding window in Simplot. Horizontal axes correspond to the relative genomic positions in K19-85-2012. (**A**) BootScan analysis shows the clustering of K19-85-2012 with other Ad-C types. The vertical axis shows the percentage of trees supporting the clustering. Series are colored according to the legend on top of the panel. (**B**) Graphical representation of the recombinant origin of the genomic segments colored as described above. (**C**) Genome annotation of K19-85-2012. (**D**) Sliding window analyses of the pairwise similarity per window of K19-85-2012 and the sequence described at the top of each chart. The vertical axis of each chart shows percentage similarity.

**Figure 5 viruses-11-01131-f005:**
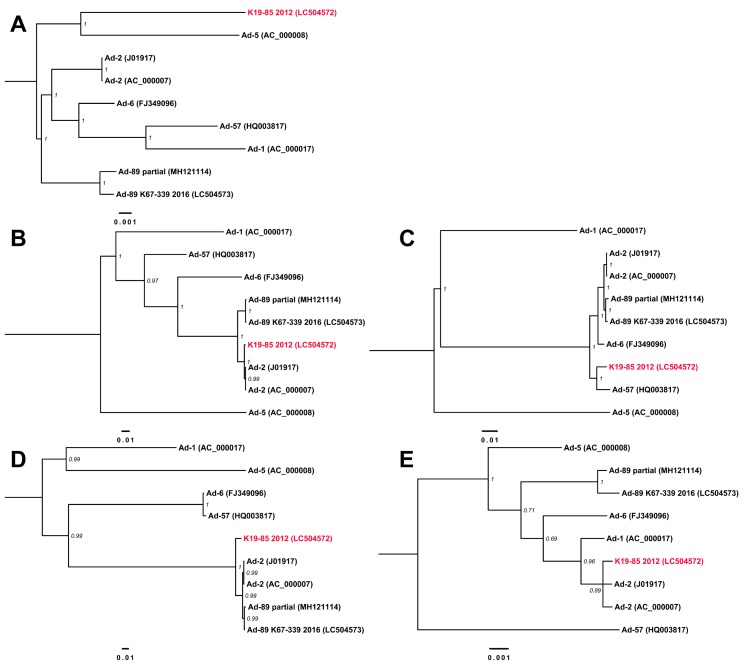
Phylogenetic trees considering the effects of detected recombination events. The Bayesian posterior probability supporting the branching is shown adjacent to the nodes. (**A**) Phylogenetic tree considering the section of the genome in K19-85-2012 detected as not recombinant with a length of 27,064 bp. Other phylogenetic trees correspond to sections of the genome for (**B**) event 1: 2527 bp, (**C**) event 2: 3685 bp, (**D**) event 3: 1527 bp, and (**E**) event 4: 1131 bp. See Table 2 and Figure 4B. Tip names of the recombinant genome sequences from the present study are colored in red.

**Table 1 viruses-11-01131-t001:** Neutralization results.

Strain *	Anti-Ad-1			Anti-Ad-2			Anti-Ad-5			Anti-Ad-6		
	5 U	10 U	20 U	5 U	10 U	20 U	5 U	10 U	20 U	5 U	10 U	20 U
K19-085-2012	++++	+++	+	-	-	-	++++	+++	+	+++	+	±
K27-108-2013	++++	+++	++	-	-	-	++++	+++	+	+++	+	-
K40-191-2014	+++	++	+	-	-	-	+++	++	+	+++	+	±
K55-255-2014	+++	++	±	-	-	-	++++	+++	+	+++	+	±
K67-339-2016	++++	+++	+	-	-	-	++++	++++	++	+++	+	±
K85-422-2017	++++	+++	+	-	-	-	++++	+++	++	+++	++	+

++++: 100%, +++: 75%, ++: 50%, +: 25%, ±: <25%, -: 0% cytopathic effect (CPE) after neutralization. * Virus isolates using A549 cells were clarified by freezing–thawing two times and centrifuging at 800× *g* for 10 min.

**Table 2 viruses-11-01131-t002:** Recombination events in Ad strain K19-85 2012 detected by the recombination detection program (RDP).

Sequence	No.	Start	End	Minor Parent	Detection Method Support (*p-*Value)						
					RDP	GENECONV	Bootscan	Maxchi	Chimaera	SiSscan	3Seq
K19-85 2012	**1**	18,725	21,251	Ad-2 (J01917)	NS	5 × 10^−9^	NS	1 × 10^−2^	1 × 10^−10^	4 × 10^−12^	1 × 10^−2^
	**2**	27,376	31,060	Ad-57 (HQ003817)	2 × 10^−31^	6 × 10^−101^	9 × 10^−100^	3 × 10^−36^	2 × 10^−32^	4 × 10^−54^	3 × 10^−15^
	**3**	31,229	32,755	Ad-2 (J01917)	5 × 10^−33^	2 × 10^−82^	5 × 10^−80^	5 × 10^−40^	2 × 10^−39^	1 × 10^−55^	3 × 10^−15^
	**4**	33,123	34,253	Ad-2 (J01917)	NS	2 × 10^−7^	2 × 10^−3^	2 × 10^−7^	1 × 10^−5^	3 × 10^−4^	2 × 10^−3^

NS: non-significant.

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
