# Peer review of "Pediatric Infections by Human mastadenovirus C Types 2, 89, and a Recombinant Type Detected in Japan between 2011 and 2018"

_viruses, 2019, doi:10.3390/v11121131_

Round 1

Reviewer 1 Report

Au have presented an adequate analysis of the genomes of species C adenoviruses isolated amonst children with respiratory infections i Tokyo. They have discovered a  new recombinant of medical importance composed of a  Ad5;pentonbase an Ad2 hexon and an Ad2fiber..   Au have  extended the knowledge  of the distribution of Ad89 to Japan and described the recombination events that can have generated Ad89.

Q: do Au think that a prior infection with   this recombinant would affect a subsequent  infection  with Ad5? ie is the pentonbase only sufficient to induce protection?

Do Au have a suggestion for a better designation  of  K19-85 2012?

line 202  "the pentonbase has been affected by multiple recombination events during the evolution of adenoviruses".Why is this so?  Which are the features of this part of the genome / the penton base protein that can accept recombination?

line 216  are purified virions or infected cell material used in the  SN assays?

Species C Ads can cause persistent shedding after infection see Kalu SU J Pedriatric Inf Dis Journal 29(8) 748-750 2010." A diagnostic conundrum? If so is the discovered  virus DNA always associated with the symptoms of the child?

Author Response

Response to Reviewer 1 Comments

Au have presented an adequate analysis of the genomes of species C adenoviruses isolated amonst children with respiratory infections i Tokyo. They have discovered a new recombinant of medical importance composed of an Ad5; pentonbase an Ad2 hexon and an Ad2 fiber. Au have extended the knowledge of the distribution of Ad89 to Japan and described the recombination events that can have generated Ad89.

Q1: do Au think that a prior infection with   this recombinant would affect a subsequent infection with Ad5? ie is the penton base only sufficient to induce protection?

Response 1

Presently, insufficient data are available to determine whether a recombinant adenovirus with a penton-base region of Ad-5 will block Ad-5 infections. However, from the children whose recombinant samples were detected during this study, Ad-5 was not subsequently detected. This epidemiological observation suggests that the recombinant form may block Ad-5 infection, however, future serological studies would be required and the protection acquired by the antibodies targeting the penton base remains to be further explored. We have added the following lines:

It also remains to be explored whether infections by these recombinant types afford any protection against Ad-2 and Ad-5 infection.

Q2: Do Au have a suggestion for a better designation of K19-85 2012?

Response 2

The designation K19-85 2012 is the strain name, and as noted by the reviewer, it corresponds to P5H2F2. In fact, we are attempting to obtain a formal number type from the Human Adenovirus Working Group. Nevertheless, the process is taking longer than expected, therefore, we opted for publishing our results to disseminate our findings while this procedure is occurring.

Q3: line 202 "the pentonbase has been affected by multiple recombination events during the evolution of adenoviruses". Why is this so?  Which are the features of this part of the genome / the penton base protein that can accept recombination?

Response 3

Our interpretation of the penton base affected by multiple recombination events comes from the evidence supporting a recombinant origin for Ad89 and the novel recombinant type of K19-85 2012. Furthermore, penton base, hexon and fiber have been characterized as recombination hotspots in multiple species of the Adenoviridae family. However, the mechanism(s) predisposing to recombination remains to be elucidated. We have added the following lines to reflect this:

In addition, considering the effects and frequency of recombination affecting the coding regions corresponding to epitope determinants in other adenoviral species, penton base, hexon and fiber have been reported as recombination hotspots [13, 36, 37], however, the mechanism(s) favoring recombination at these loci remains to be elucidated.

Q4: line 216 are purified virions or infected cell material used in the SN assays?

Response 4

We modified the footnote of the Table 1 as follows to clarify:

*Virus isolates using A549 cells were clarified by freezing-thawing two times and centrifuging at 800×g for 10min

Q5: Species C Ads can cause persistent shedding after infection see Kalu SU J Pedriatric Inf Dis Journal 29(8) 748-750 2010." A diagnostic conundrum? If so is the discovered virus DNA always associated with the symptoms of the child?

Response 5

We thank the reviewer for the important observation and agree with the possibility of virus shedding despite no relationship to the exhibited symptoms; furthermore, we have added the reference and noted this limitation as part of the study. However, association of Ad types and symptoms in the observed cases were supported by at least three factors: (1) the symptoms were taken from multiple cases; (2) other types of Ads were not detected in the patients, and (3) the high concentration of viral particles per sample to be detectable by immunochromatography kits suggests active infections. These factors were added as follows:

It is noteworthy that although Ad-C infections can cause persistent infections with viral shedding that could limit the determination of a definitive diagnosis of the pathogen eliciting the exhibited symptoms [39], the association of symptoms upon clinical presentation and the Ad types were supported by three factors: (i) the symptoms were summarized from multiple cases; (ii) other types of Ads were not detected in the infections, and (iii) the high concentrations of viral particles per sample needed to be detectable by immunochromatography kits suggests active infections.

Reviewer 2 Report

This is a very sound paper that documents in detail the molecular characteristics of adenoviruses identified in a Japanese paediatric clinic over many years. Most of these viruses are based on species c, type 2 and there are interesting novel features of some of these viruses that are certainly worth reporting. The work extends our understanding of natural infections of adenoviruses in children and the evolution of these viruses. 

Author Response

Thank you for your review.